# Evaluation of Multispectral Data Acquired from UAV Platform in Olive Orchard

**Pietro Catania** , **Eliseo Roma \*** , **Santo Orlando** and **Mariangela Vallone**

Department of Agricultural, Food and Forest Sciences, University of Palermo, 90128 Palermo, Italy

\* Correspondence: eliseo.roma@unipa.it

**Abstract:** Precision agriculture is a management strategy to improve resource efficiency, production, quality, profitability and sustainability of the crops. In recent years, olive tree management is increasingly focused on determining the correct health status of the plants in order to distribute the main resource using different technologies. In the olive grove, the focus is often on the use of multispectral information from UAVs (Unmanned Aerial Vehicle), but it is not known how important spectral and biometric information actually is for the agronomic management of the olive grove. The aim of this study was to investigate the ability of multispectral data acquired from a UAV platform to predict nutritional status, biometric characteristics, vegetative condition and production of olive orchard as tool to DSS. Data were collected on vegetative characteristics closely related to vigour such as trunk cross-sectional area (TCSA), Nitrogen concentration of the leaves, canopy area and canopy volume. The production was collected for each plant to create an accurate yield map. The flight was carried out with a UAV equipped with a multispectral camera, at an altitude of 50 m and with RTK correction. The flight made it possible to determine the biometric condition and the spectral features through the normalized difference vegetation index (NDVI). The NDVI map allowed to determine the canopy area. The Structure for Motion (SfM) algorithm allow to determine the 3D canopy volume. The experiment showed that the NDVI was able to determine with high accuracy the vegetative characteristic as canopy area (r = 0.87 \*\*\*), TCSA (r = 0.58 \*\*\*) and production (r = 0.63 \*\*\*). The vegetative parameters are closely correlated with the production, especially the canopy area (r = 0.75 \*\*\*). Data clustering showed that the production of individual plants is closely dependent on leaf nitrogen concentration and vigour status.

**Keywords:** DSS; NDVI; precision oliviculture; remote sensing

## 1. Introduction

In recent years, there has been an increase in olive growing and in the consumption of extra virgin olive oil (EVOO, [1]). It is cultivated almost entirely (over 98%) in countries of the Mediterranean area where traditional agronomic practices are used. However, a continuous change of landscape and cultivation techniques has been observed and requires appropriate agronomic choices for a successful crop. This situation, poses new challenges to ensure environmental and economic sustainability of olive farms [2]. These agronomic techniques are able to modify the vegetative and productive activity of the olive tree and require appropriate choices depending on the agro-climatic context (phytosanitary management, irrigation, soil management, pruning, fertilization, etc . . . ). Instead, the management of each single plant may depend on various conditions and thus requires differentiated management practices.

Since 1990's precision agriculture (PA) gave the farmer the possibility of changing crop management. Indeed, PA is a strategy of management which takes into account variability with the goal of increase crop efficiency and production quality and quantity. Variability can be expressed in several ways. As reported by Zhang et al., (2002) [3], variability affecting agricultural production can be classified into six groups: yield variability; field

variability; soil variability; crop variability; variability related to abnormal factors; and management variability. Moreover, there are various sensors and platforms which can be used to investigate variability [4–6]. Senay et al., (1998) [5] distinguish three ways of measuring spatial variability in the field: continuously, discretely (e.g., point sampling of soil or plant properties), and remotely (e.g., through aerial photographs). Discrete sampling is generally characterized by a high precision of the investigated variable but cannot describe the complete variability and need precision geo-statistic techniques to spatialise the data [7–9]. Therefore, remote sensing represents the most important technique of acquisition data for olive growth in precision agriculture management [4,10]. This, can be performed using platforms at different distance from the object, which determines the area wideness [11,12]. Precisely, in the olive orchard the most important platform used is the Unmanned Aerial Vehicle (UAV, [4,6]) because it allows to determine huge areas in a low time of flight and can be equipped with different sensors [13,14].

In olive orchards traditionally managed, fertilizers and other inputs are applied at uniform rates without considering the field spatial variability [15]. This management may result in under-application or over-application of inputs with obvious economic and environmental problems [16,17]. Furthermore, the abuse of the main agronomic source as fertilizers and water can compromise the quantity and the quality of the production [18,19]. Therefore, it is important to know the spatial variability of soil, crop and climate in order to apply the best site-specific management and to improve economic and environmental sustainability in olive orchard. Soil variability is probably able to determine a more general state of fertility of the entire agro-ecosystem [7,20,21] while, the climate variability source is low modifiable. For this reason, it is better to investigate directly the crop characteristics.

The crop health status can be observed from several crop traits such as: nutritional [22,23], structural-biophysical [24,25], spectral [26], and productive [21,27]. The structural-biophysical status is strictly related to the vigour behaviour and can be measured in several ways such as TCSA (trunk cross section area) [9,26], LAI (Leaf Area Index) [24] or canopy volume [25].

The nutritional status is investigated using the analysis of leaves, as made by López-Granados et al., (2004) [22] who created a site-specific fertilization map for olive trees based on leaf nutrient spatial variability.

The knowledge of the biophysical characteristics of the plants is being very successful, in the last years, because it can be estimated using different sensors [28] such as LiDAR [29], low cost RGB cameras [30], and other, with high correlation with the spatial health condition of the olive tree [31]. If the real conditions of the foliage volume were known precisely, it was possible to better regulate some treatments associated with it, such as phytosanitary treatments, obtaining considerable savings in economic and environmental terms [17,32]. The production of each plant is a good indicator of health status but can be determined only at the end of the year and can depend on other parameters [33].

Since the beginning of PA, the use of multi and hyperspectral information from the crop has increased because spectral information's are closely related to the health status of each tree. This information can be obtained through the use of Vegetation Index (VI),widely applied in the olive grove [4,34,35]. It is able to investigate with high precision a huge area and the main vegetative characteristics that are closely related with productivity. Furthermore, by succeeding in modifying the vegetative characteristics, it is possible to achieve a vegetative-productive balance and maximum efficiency of the agro-ecosystem.

Recent advances in modelling and decision support systems (DSS) applied to agriculture promise to bring about important positive changes in olive orchard management. In order to be applied in olive grove, they require a high level of specific information providing a good understanding of the growing conditions of the plants [36,37]. In the literature, several studies have investigated the potential of the new technologies proposed for intelligent agriculture on the determination of certain crop parameters. Therefore the agricultural sector needs good indicators to accurately and reliably analyse multispectral plant information in order to be applied in precision agriculture using DSS [38–40]. The

olive growing sector needs information on the agronomic practices to be applied during the growing cycle to optimize crop management.

Based on the above, the aim of this study was to investigate the ability of multispectral data acquired from a UAV platform to predict nutritional status, biometric characteristics, vegetative condition and production of olive orchard as tool to DSS.

## 2. Materials and Methods

### 2.1. Study Area

The study area is located in Calatafimi Segesta (Trapani, Italy); it has a surface of 5860 m$^2$ and a perimeter of 344 m with flat orography (Figure 1). According to the Koppen–Geiger's classification, the climate of the area is classified as Csa (Mediterranean hot summer climates; [41]). Climatic data of the year show a mean annual air temperature ranging from 18 °C to 22 °C and a mean annual precipitation of 550 mm (Sicilian Agrometeorological Information Service). The soil moisture regime is xeric, border with the aridic one, and the temperature regime is thermic.

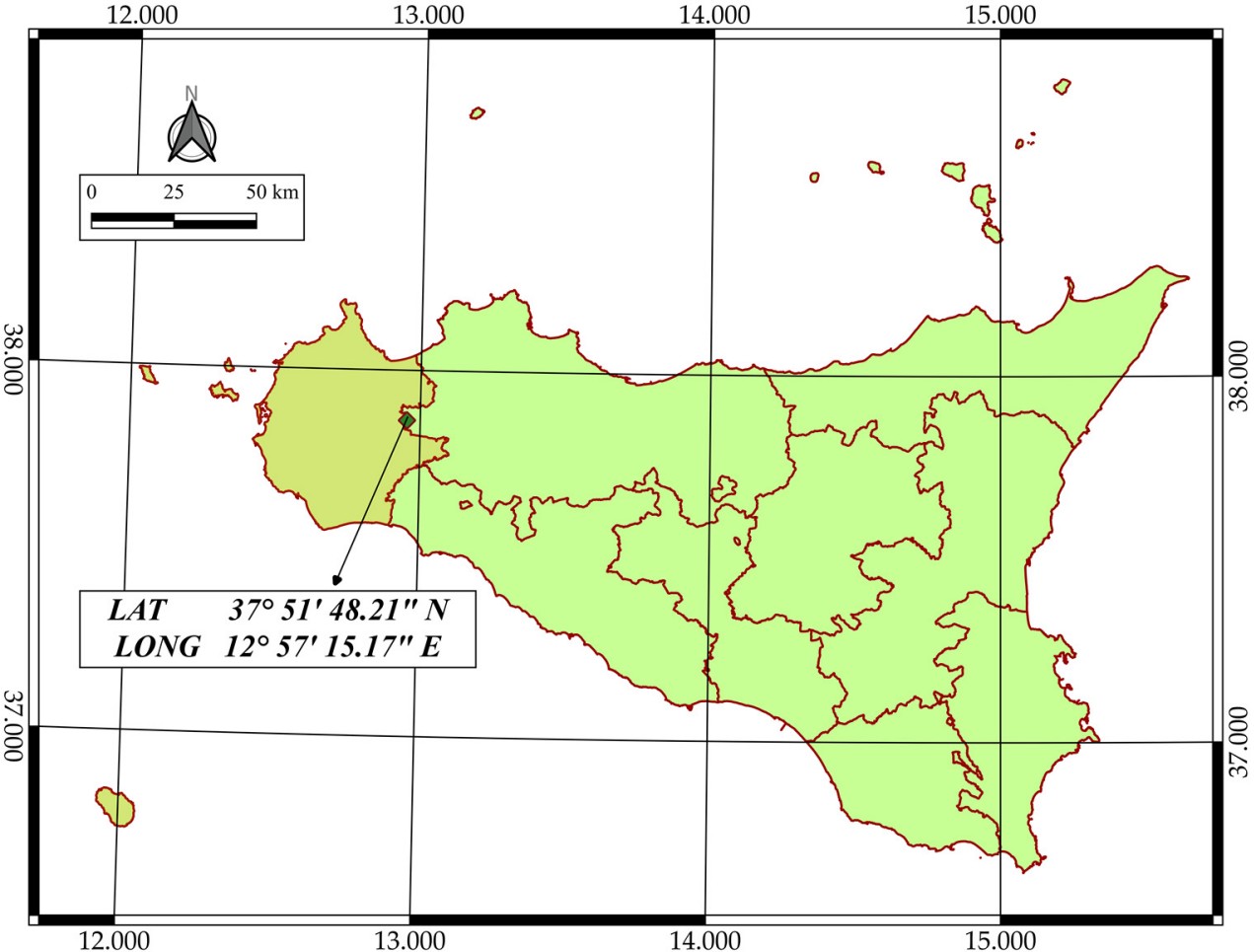

**Figure 1.** Geographical position of the study area in Sicily (Italy).

The study was carried out during the 2021 crop season in an olive orchard managed with ordinary practices in rainfall system. The olive grove, cv. Cerasuola, was in full productivity at the time of the experimentation. The plot layout has traditional training system with distance of 5 m × 5 m; the total number of trees considered in the tests was 211 (Figure 2).

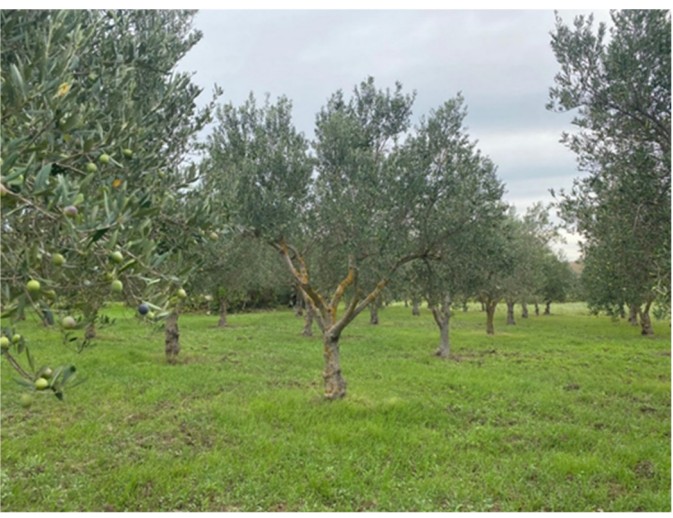

**Figure 2.** Experimental olive grove.

Figure 3 shows the flowchart of the methods used for data acquisition and processing.

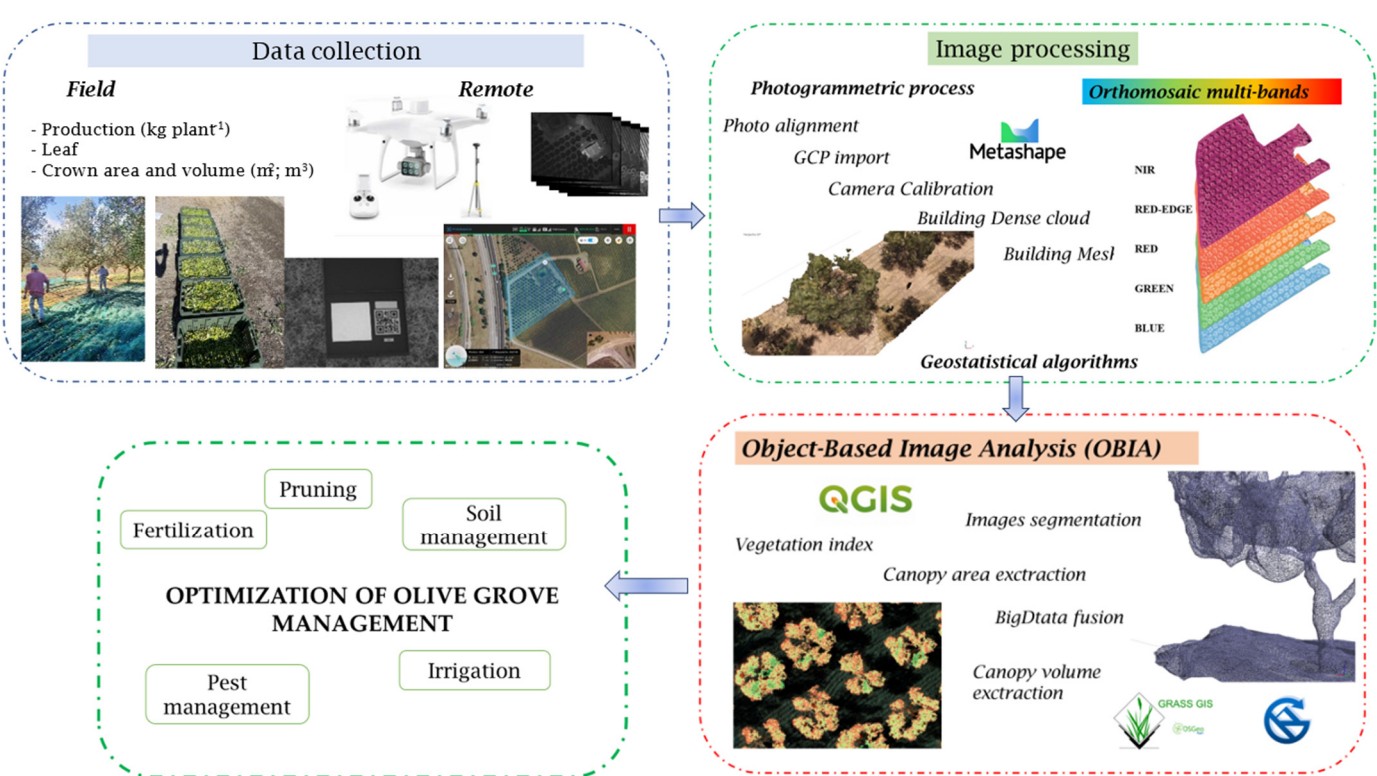

**Figure 3.** Overall flowchart of the experimental process.

### 2.2. Field Data Collection

Plot perimeter and plants were georeferenced on DOY (day of years) 161 using the instrument Stonex S7-G (S7-G, StoneX, Paderno Dugnano, Italy) with differential RTK (Real Time Kinematic) correction as used in other studies to have a good accuracy and precision [42,43]. This instrument is able to receive L1 (1575.42 MHz) and L2 (1227.60 MHz) frequencies of the main constellation and it is also equipped with a slot for a SIM card and a GSM/GPRS/EDGE modem, in order to obtain real-time differential correction data from the RTK ground station network (CORS). On DOY 163 the TCSA at 0.50 m from the ground (trying to exclude any hyperplasic nodes typical of the olive tree) was measured for all

plants [9]. On day 164, 50 trees were randomly selected from the field in which the height and diameter of the crown were measured manually with a ruler.

Field samplings were carried out in order to investigate nutritional status using a regular 15 m × 11 m grid on DOY 205 [22]. The sampling point was identified at the intersection point (node) of the sampling grid, excluding the most external part of the field (Figure 4). A total number of 36 points was sampled. The sample was represented by an experimental unit of four adjacent olive trees. Each leaf sample consisted of four sub-samples of 25 healthy, fully expanded and mature leaves, collected from the central portion of the current season's unshaded branches at a height of 1.5 m above the ground surface, at the four cardinal points of each olive tree.

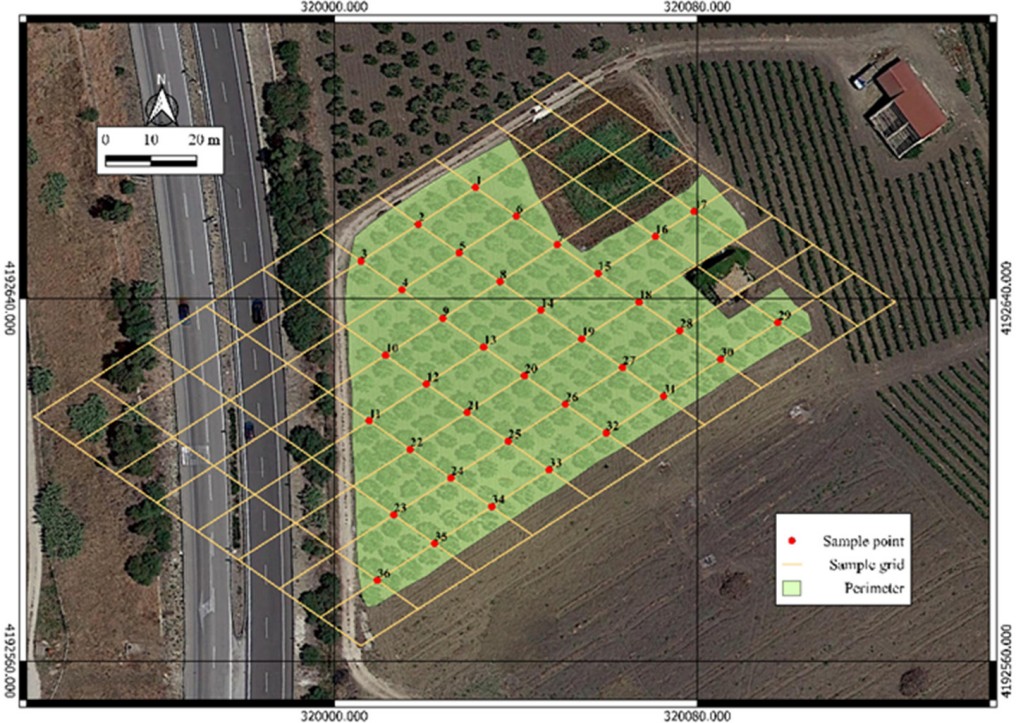

**Figure 4.** Sample grid and sample points used for nutritional and soil condition characterization.

After sampling, the leaves were dried at 70 °C for 24 h and milled to pass through a 0.25 mm mesh. Leaf samples were analyzed to determine the total nitrogen content (N) by the Kjeldahl method.

The olives were picked with a hand-held electric harvester model OLIVION P230 (Pellenc, SI, Italy; Figure 5), when their maturity index was equal to 2.38, determined according to Furferi et al., (2010) [44]. Two operators had the task of laying and wrapping the nets under each plant. Finally, the production of each plant and that of the whole plot were evaluated quantifying the harvested olives using a proper load cell [45].

### 2.3. Multispectral Data from UAV and Flight Scheduling

Multispectral data were acquired through an aerial survey using a Phantom4 Multispectral (DJI, Shenzhen, China). The Unmanned Aerial Vehicle (UAV) is equipped with four rotors on a rotating wing, one brightness sensor at the top. It is also capable of image position compensation as the relative positions of the CMOS sensor centers of the six cameras and the phase center of the on-board D-RTK antenna, are stored in the Exif information of each image. The multi-frequency Global Navigation Satellite System (GNSS) positioning system can see and receive signals from the main constellations.

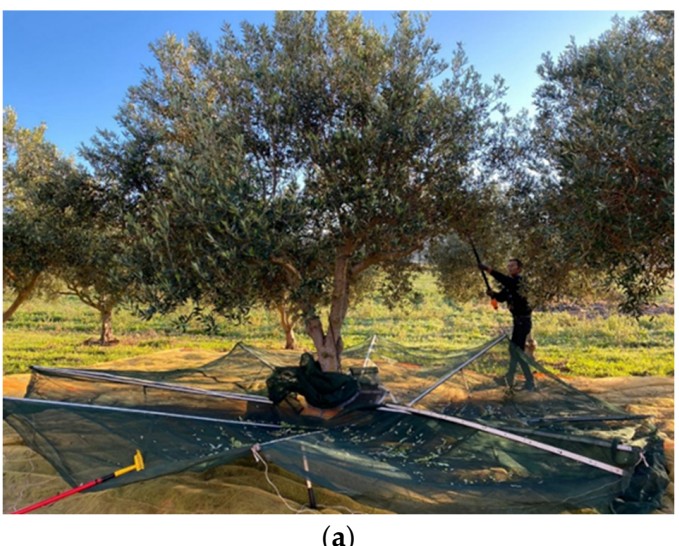

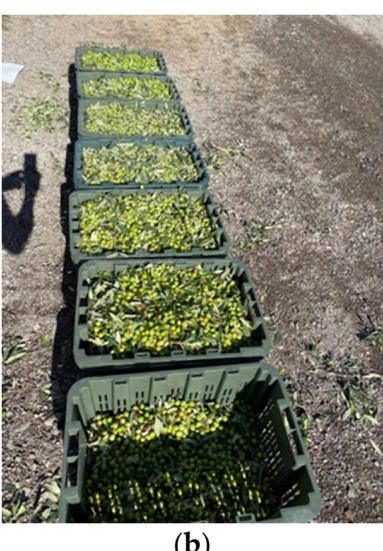

(**a**)          (**b**)

**Figure 5.** Field use of the Pellenc Olivion P230 shaker (**a**). Harvested olives (**b**).

The multispectral camera has six 1/2.9″ CMOS sensors, that is an RGB sensor for visible light imaging and five monochrome sensors for multispectral imaging with a final resolution of 2.08 MP pixels. The monochromatic bands are Blue (B), Green (G), Red (R), Red-Edge (RE) and Near InfraRed (NIR), respectively with the following central wavelengths: 450 nm, 560 nm, 650 nm, 730 nm and 840 nm. The spectral resolution for R, G, B, RE bands is $\pm 16$ nm and $\pm 26$ nm for the NIR band. The lens has 62.7° FoV (Field of View), 5.74 mm focal length and f/2.2 aperture.

The flight was conducted in 2021 with automatic configuration using the waypoints and RTK mode for correcting geospatial data. The flight was performed at approximately 12:00 noon at a flight of 50 m, generating a ground surface distance (GSD) of 2.6 cm. Five GCPs (Ground Control Point) were placed before the flight. The GCPs were georeferenced using the Stonex S7-G instrument with an external dual-frequency antenna (L1/L2; Stonex geodetic antenna) in RTK mode and averaging about 60 coordinate points. Image acquisition was made at an average speed of 10 m s$^{-1}$ in stop-and-go mode to minimize speed-related distortions. Both front overlap ratio and side overlap ratio were 70% while the gimbal pitch was set at 90° (downwards).

*2.4. Image Processing*

The photogrammetric reconstruction was carried out using Agisoft Photoscan Professional version 1.7.3. The photogrammetric process employed is the classic scheme to reconstruct the orthomosaic multi-bands. Precisely, the different band image has been downloaded and uploaded in the software. The next steps were alignments, GCP upload, calibration in reflectance. Once the preparation was complete, the dens cloud, the Digital Elevation Model (DEM) and finally the orthomosaic multiband were constructed.

For geo-spatial data analysis and processing the open-source software QGIS ver. 3.2 [46] was used. The main geostatistical methods were used to create the various maps that allowed to compare and analyze the variability found in the olive orchard. Through a process of rasterization and vectorialization algorithm, it was possible to extract different information on the tree ([24]; Figure 6). The canopy area (CA) and crown volume (CV), were the bio-metric information while the spectral information was derived from the calculation of the main vegetation index (VI) used in the literature to determine vigour characteristics [4]. The VI used was the Normalized Difference Vegetation Index (NDVI; [34]) that was calculated using Equation (1).

$$NDVI = \frac{\rho_{Nir} - \rho_{red}}{\rho_{Nir} + \rho_{red}} \qquad (1)$$

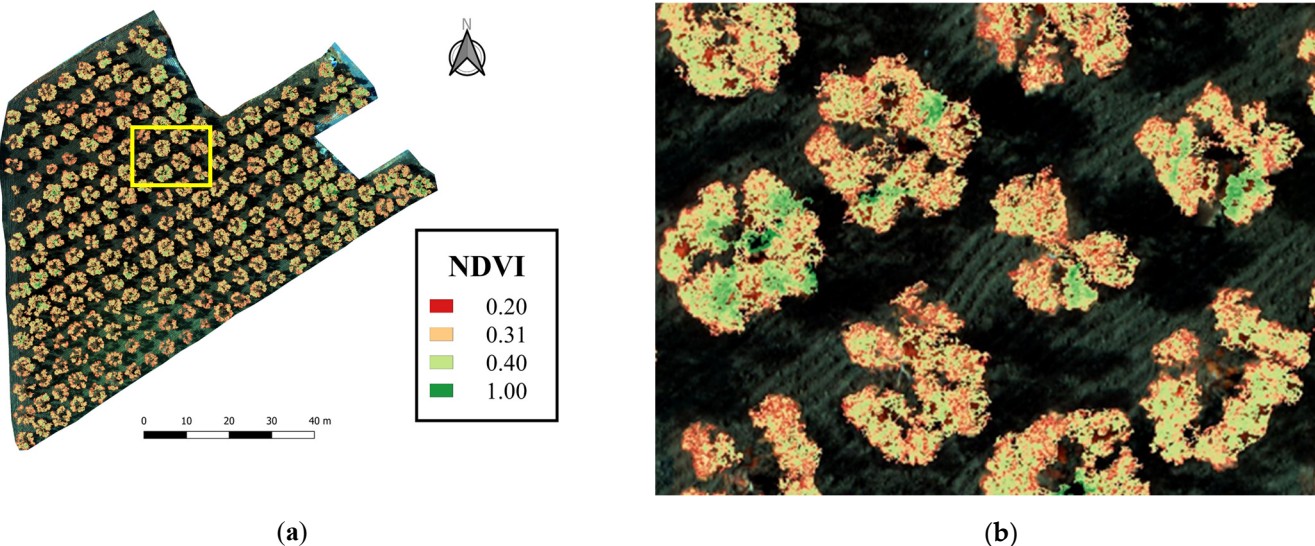

(**a**)　　　　　　　　　　　　　　　　　　　　　　　　(**b**)

**Figure 6.** (**a**) NDVI for canopy and false background color image. (**b**) Zoom of canopy extracted for plant and corresponding NDVI.

CA was extracted starting from the NDVI map among several OBIA steps. The first step was the image segmentation to differentiate the canopy from the background. It was performed using the K-means algorithm executed in a Saga tool of raster image analysis.

The DSM was extracted directly from the photogrammetric processing while the DTM was derived from some terrain point random selected and spatialized using a geostatistical method (Figure 7). After calculating DTM and DSM, it was possible to determine the CV using the following Equation (2) as defined in [24]:

$$CV = (DSM - DTM) - TrH \tag{2}$$

where DSM is the Digital Surface Model; DTM is the Digital Terrain Model; TrH is the trunk height (mean value of the 50 selected plants).

### 2.5. Biometric Data Analysis

The aptitude of the orthomosaic and their DSMs to build the tree structures and to retrieve their geometric features was evaluated. These parameters are namely projected area of the canopy (CA) and crown volume (CV); they were evaluated by comparing the UAV-estimated values and the on-ground values observed in the validation fields. In the case of the CA, the same methodology was applied in Torres-Sánchez et al., (2015) [25] in order to better quantify this variable. For this purpose, fifty olive trees were randomly selected in the field and their shape was outlined manually using the orthomosaic image to be used as an observed measure. The results of the GEOBIA (geographic object-based image analysis) analysis on the estimation of the CA and CV were compared to the observed measures to calculate the area of coincidence for each olive tree and calculate the overall accuracy. In the same olive trees selected for CA, the CV quantification and validation were applied. CV* was estimated starting from the manual measurement, assuming an ellipsoid form and applying a validated method (Equation (3)) for olive tree geometric measurements using the parameters measured on DOY 164 [25,47].

$$CV^* = \frac{\pi}{6} * \left( \frac{C_l * C_w}{2} \right)^2 * \frac{T_h}{2} \tag{3}$$

where $C_l$ is the Canopy length (m); $C_w$ is the Canopy width (m); $T_h$ is the tree canopy height (m). The effectiveness of the entire procedure to measure volume and area of the

canopy of the fifty selected trees was evaluated by calculating the root mean square error (RMSE) and correlation coefficient derived from the regression fit.

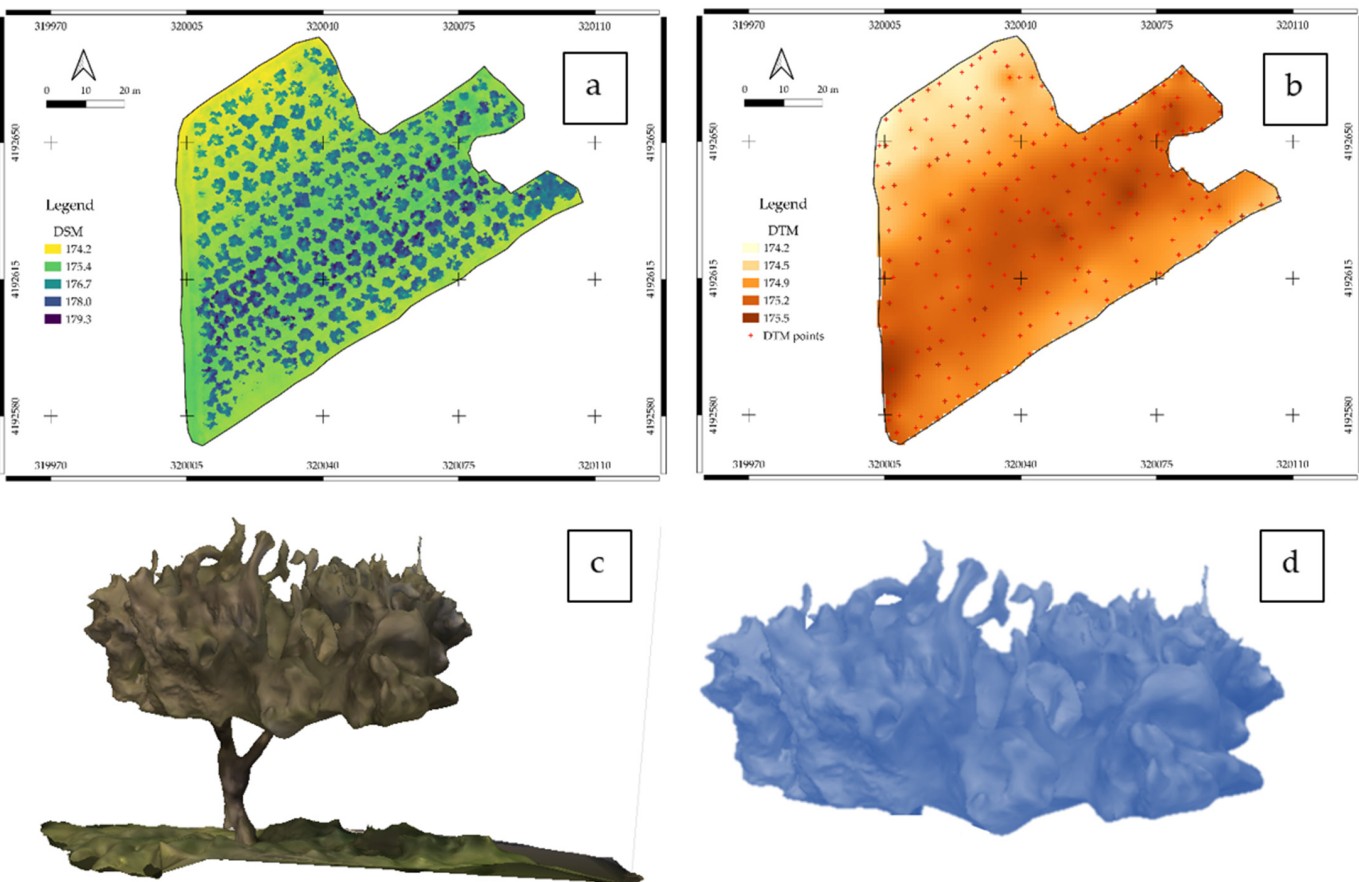

**Figure 7.** Data processing steps to obtain CV. (**a**) Digital Surface Model (DSM); (**b**) Digital Terrain Model (DTM); (**c**) Representation of the tree; (**d**) Crown volume obtained using Equation (1).

## 3. Results

In our experiment we evaluated the nutritional, spectral, vegetative and production spatial variability in the olive grove. With regard to the crop nutritional status, it was investigated only TN, that showed a concentration below the threshold in all the samples as obtained by other authors [48–50]. Indeed, the total N concentration of plant leaves ranged from 0.4% to 1.46%, with a mean value of 0.92%. By geostatistical analyzing the maps, it was possible to obtain TN spatial variability.

Regarding the vigor characteristics, such as TCSA, a certain heterogeneity among the plants in the field was observed. The mean TCSA value of the whole plot was 297.3 cm$^2$ ± 109.6. These differences were reflected in growth and production activity as showed also in Noori and Panda (2016). Indeed, the TCSA values were statistically significant correlated with different variables expressing plant vigour such as canopy area extracted from the multispectral image (r = 0.65 ***; Figure 8a). TCSA also statistically significant correlated with NDVI (r = 0.58 ***; Figure 8b) and productivity values (r = 0.42 ***, data not show).

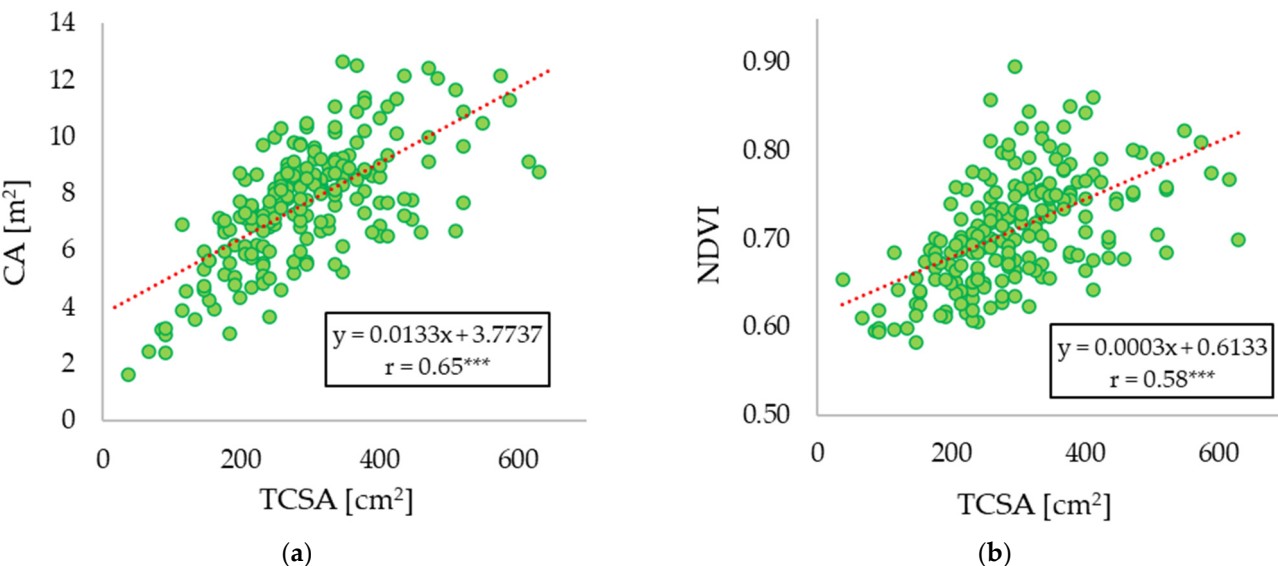

**Figure 8.** (**a**) Correlations between TCSA and CA; (**b**) correlations between TCSA and NDVI calculated as the average of all pixels within the CA of each tree. *p* value < 0.001 (***).

NDVI, CA and CV have been calculated using the drone's multispectral image and GIS processing; therefore, they made it possible to quickly and easily investigate the variability of the field. NDVI, CA and CV had respectively an average value of $0.71 \pm 0.06$, $7.7 \pm 2.09$ m$^2$ and $18.02 \pm 2.2$m$^3$. Crossing all the vigour parameters such as CA, CV, and TCSA, the plants were clustered in three vigour groups (C1, C2, C3) using K-means as cluster algorithm. These cluster groups represent the three-vigour classes: High (HV), Medium (MV) and Low Vigour (LV).

The three vigour groups showed clear differences in terms of vigour (Figure 9). The three parameters showed an increasing data trend for the three vigor groups. CA showed values of 5.4 m$^2$ $\pm$ 0.8, 8.15 m$^2$ $\pm$ 0.6 and 9.6 m$^2$ $\pm$ 0.65 for the three-vigour levels, respectively; NDVI showed values of $0.64 \pm 0.02$, $0.72 \pm 0.02$ and $0.78 \pm 0.02$ going from C1 to C3; CV showed values of 15.6 m$^3$ $\pm$ 0.84, 18.5 m$^3$ $\pm$ 0.73 and 20.1 m$^3$ $\pm$ 0.80 for the three-vigour levels. From the statistical analysis, it appears that the NDVI of each individual tree was able to describe the variability of the field especially in terms of vigour characteristics. In fact, NDVI statistically significant correlated with the values of canopy area (r = 0.87 ***, Figure 10a). Furthermore, the NDVI showed a good relationship with production activity (r = 0.63 ***, Figure 10b).

Also CA had a good influence on the productivity of the olive grove. Indeed, it was observed that productivity depends on the canopy area of the single plants (r = 0.75 ***, Figure 11). This result is supported by PCA analysis, where it was possible identify as the trees with high and low vigour were clustered with high and low production respectively (Figure 12). The average production and CA of all plants were used as a threshold to distinguish high and low production and canopy area.

The application of image reconstruction using SfM techniques allowed the generation of detailed DSM, DTM and orthomosaic, as shown in Figure 13. CV showed a good ability in reconstructing the geometry for each individual tree in the whole plot. Indeed, it showed a strong relation with the other vigour parameter and with the production capacity of the plants (r = 0.74 ***).

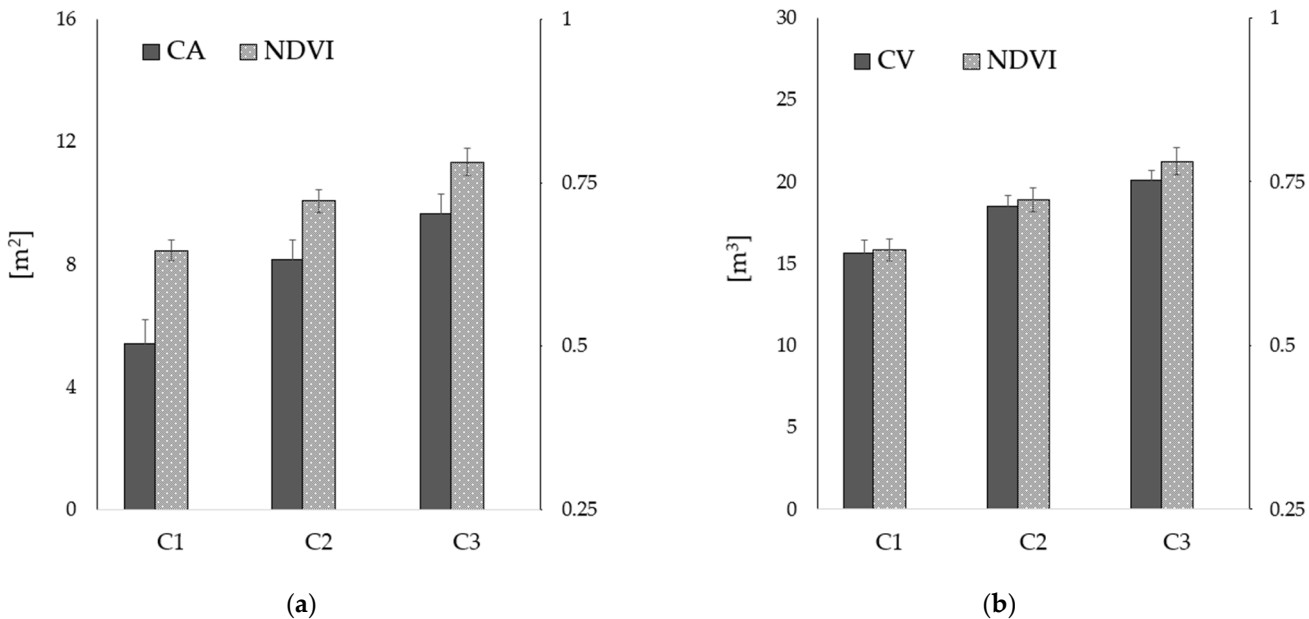

**Figure 9.** (**a**) CA and NDVI values (±st. dev) for the three clusters; (**b**) CV and NDVI values (±st. dev) for the three clusters.

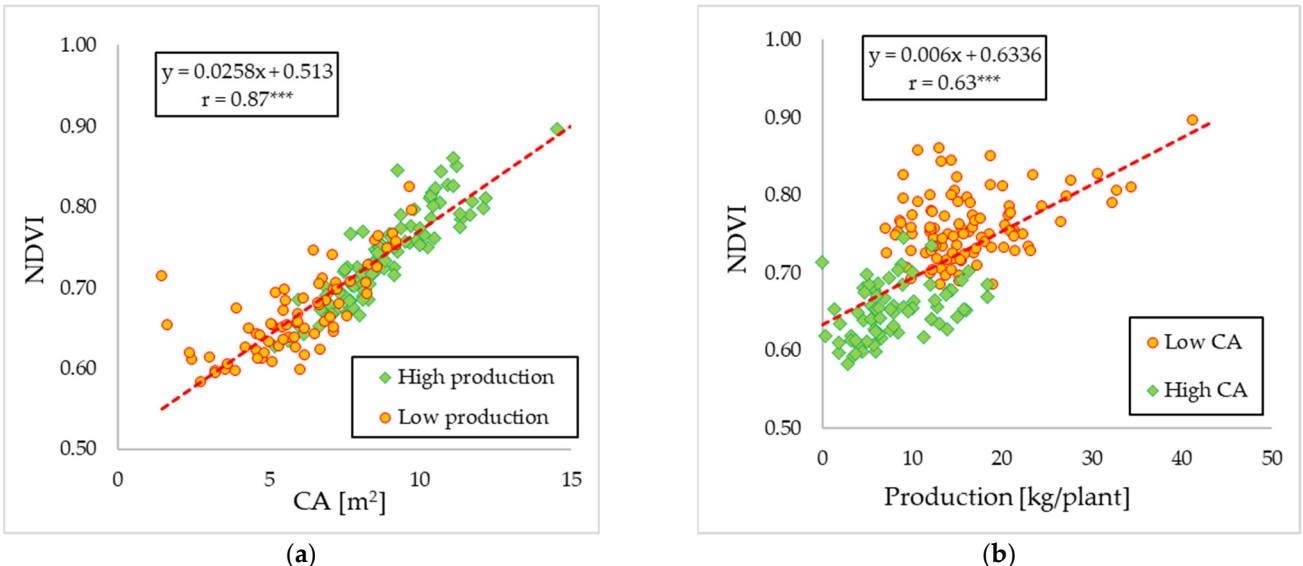

**Figure 10.** (**a**) Correlation value between NDVI and Canopy area (m²); (**b**) NDVI and Production (kg/plant).

Starting from the CA and CV calculated using the Qgis software, it was possible to carry out data validation using the ground truth with a good accuracy. In fact, the accuracy assessment between the observed and estimated values for CA resulted in RMSE equal to 0.54 and a statistically significant close linear relationship with $R^2 = 0.98$ ***.

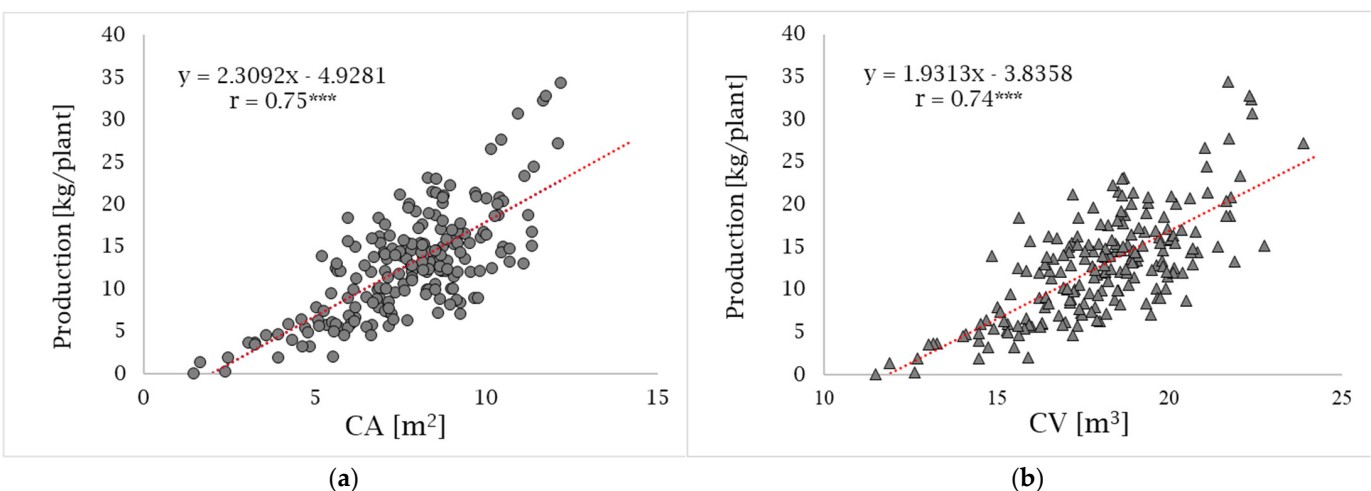

**Figure 11.** (**a**) Correlation between CA and Production; (**b**) correlation between CV and Production.

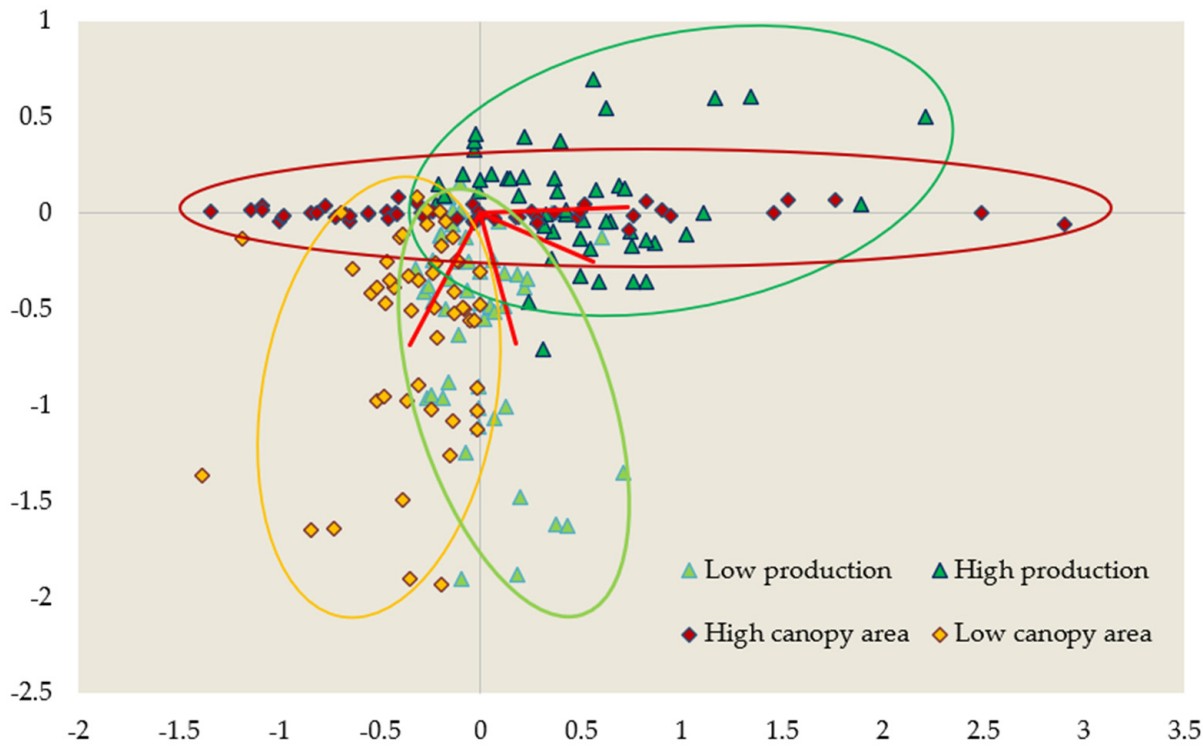

**Figure 12.** Principal Component Analysis (PCA) of high and low CA and production.

CV showed an underestimation of the final volume when compared to field measurements. In this case, the coefficient of determination was $R^2 = 0.67$ *** with RMSE equal to 9.5 m$^3$ (Figure 14). Volume differences between the observed and estimated values do not denote real errors of the UAV-based measurements because the ground-based values were derived by applying the geometric equation that considers trees as full, ellipsoid shapes producing inaccurate estimates [25,51]. In contrast, the three-dimensional products derived from the 3D reconstruction, reproduce the irregular shape of the canopy, yielding better estimates of tree volume as showed in Figure 15.

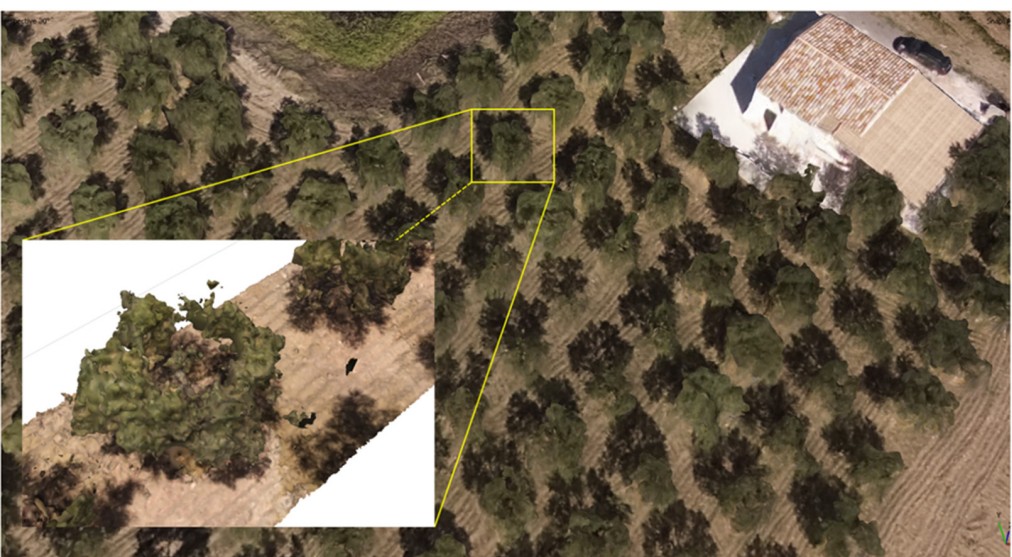

**Figure 13.** 3D reconstruction of the whole study area and zoom of one tree.

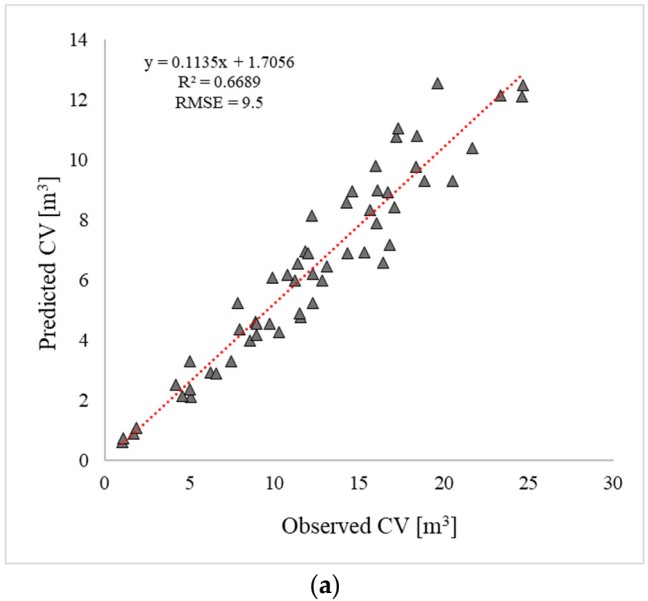

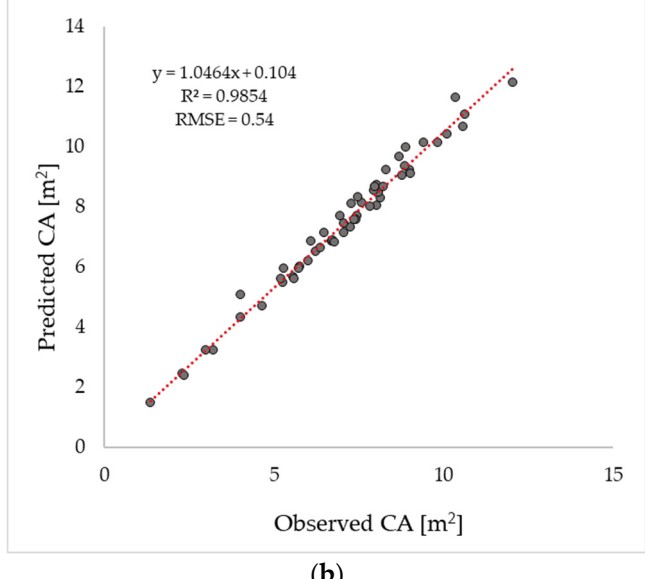

| (**a**) | (**b**) |

**Figure 14.** (**a**) Comparison between ground measured and UAV-estimated CV; (**b**) comparison between ground measured and UAV-estimated CA.

Starting from the leaves sampling, it was possible determine the crop nitrogen status. The mean nitrogen concentration of the whole plot was 0.94 % depending of the tree and influenced only by the production activity (data not show), while the vegetative characteristic wasn't correlated with it. It was also possible to cross the different information with GIS program because yield depends on vegetative and nutritional status. Using the three groups of cluster and plotting their score of nitrogen concentration and canopy area, it was observed that the whole plot showed clear heterogeneities. These clusters were statistically different *p* (<0.001) in terms of productivity by ANOVA analysis (Figure 16). Moreover, the ANOVA test showed that CA has a greater effect than nitrogen concentration.

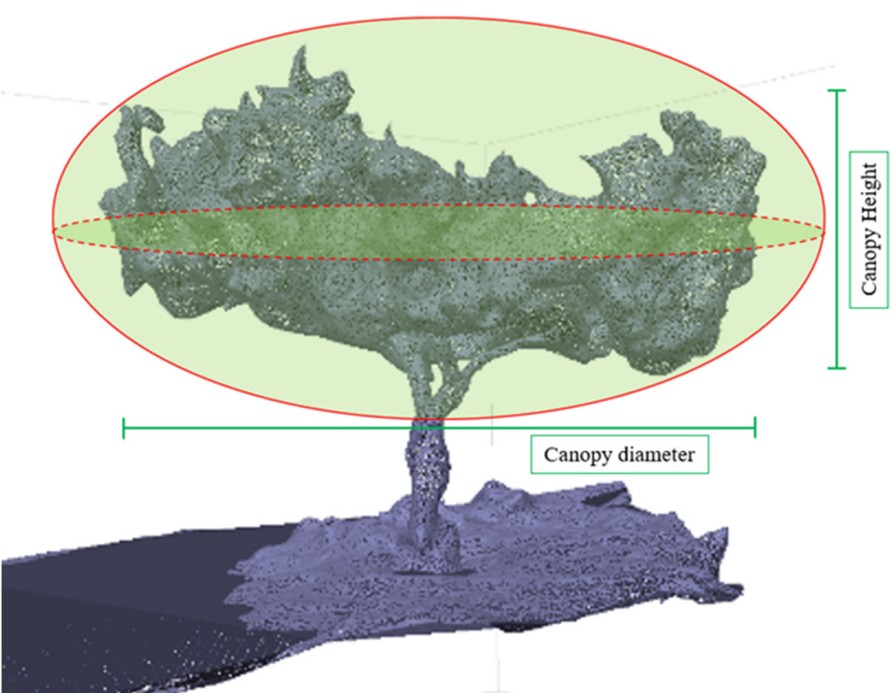

**Figure 15.** Comparison between geometric (red shape) and SFM reconstruction of the tree.

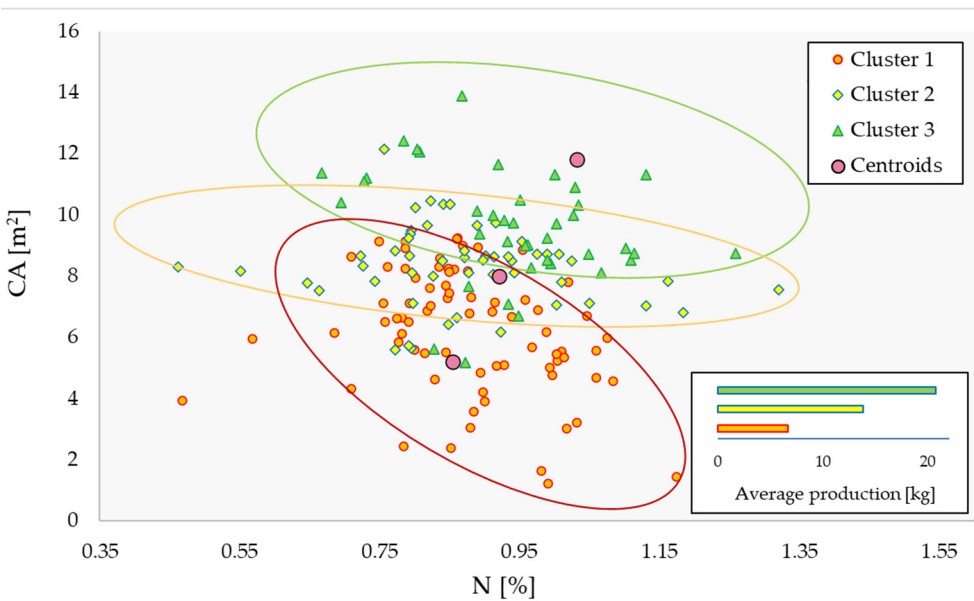

**Figure 16.** Cluster analysis of the three vigour groups (C1, C2, C3) according to total nitrogen content and CA and ANOVA test results for the production.

## 4. Discussion

The technologies available today in precision farming are able to describe and determine the health status of the olive grove. Variability can be observed both in terms of soil and cultivation characteristics but the last ones are the most important to investigate the variability as showed in other studies [24,52]. Indeed, in our study the crop health status was determined using the vegetative, spectral and productive activity of individual trees. As confirmed by different studies, these parameters are strictly related and their knowledge can be used to opportunely manage the orchard [21,24,53]. As previously mentioned, the plant health was the most important factor in determining the production result. In the

present work it was expressed from spectral, biometric and nutritional point of views. In general, the nutritional status of all the plants was deficient, as the whole plot had a foliar nitrogen concentration below the minimum threshold. Probably, the nutritional status was the main limiting factor for plant growth in the considered plot as found in another study [7]. The low NDVI values found [9,22] and the entire vegetative heterogeneity detected could be explained by the deficient nutritional status of the plants. Indeed, the nutritional condition for each plant was not correlated well with the vegetative parameters. However, this deficiency was one of the determining factors for plant production as obtained by PCA that supported the plant clustering into three groups (Figure 16).

To express the vegetative variable, i.e., vigour, it was decided to use the TCSA, the canopy projection area and the canopy volume. Since the characteristics express a condition of vegetative vigour, all variables showed at least moderate correlations (Figure 8). TCSA is a condition formed over the years of cultivation and it cannot describe the annual condition of the plant, while the area of the canopy certainly expresses a precise condition at an exact moment. Probably, for this reason the canopy area was indeed more correlated with plant production and NDVI (Figure 10). The crop spectral conditions were investigated by calculating NDVI that describes the general vegetative and nutritional conditions of each plant because the bands used for its calculation (NIR and red) are strongly related with them [54,55]. In this study, the NDVI showed low values, especially where conditions of low vigour and low nitrogen concentration in the leaves were found. As also show in other studies [24,56] the NDVI has a good relation with the vegetative status (Figure 10). Moreover, when it correlates with the CA, it was able to discern the plants with high or low productivity with good precision (Figure 10a). When it was correlated with production, it was able to underline the plant with high vigour (precisely with high canopy area; (Figure 10b). NDVI showed better correlations with canopy area than the vigour parameters because the multispectral bands used in the calculation are sensitive to both effects: leaf efficiency (red band) and canopy structural conditions (NIR band) [54]. Since production was mainly linked to the availability of plant resources and therefore to CA, NDVI always proved to be a good indicator and predictor of production even in non-optimal nutritional conditions. These results emphasize that NDVI is more capable to determine the vegetative parameters than production. Therefore, by having precise multispectral and RGB images of the entire olive orchard, it is possible to use this information to obtain crop status data that can be used in development models or DSS for the optimization of agronomic management.

Crossing the spectral, biometric, productive and nutritional characteristics of each plant by cluster analysis very interesting results appear. Three statistically different clusters (C1, C2, C3) were identified by cluster analysis according to their vigour and nutrient characteristics (Figures 9 and 16). The production of the three clusters showed statistically significant differences. C3 was the most productive and vigorous, while C1 was the lowest (Figure 9). It shows that the productivity of the plants is positively related to the development of the canopy and secondarily to the nutritional conditions. High productivity was observed for plants with a very vigorous canopy and discrete foliar nitrogen concentrations. These results confirm that vegetative conditions were the main determinants of production, while nutritional status had no effect. These results are also supported by PCA (Figure 16). Indeed, plants with high production and CA are classified as a more similar group than those with low production and CA. This effect can be explained by the greater availability of accumulated resources in the reserve organs of the more vigorous plants.

The UAV equipped with multispectral and RGB camera showed a good capacity to extract the vegetative information using spectral and biometric data. They can be able to predict the production and consequently to better manage variability with significant environmental, agronomic and economic benefits [17,20]. Geometric reconstruction showed interesting results. The high value of RMSE obtained between observed and estimated data were found in previous studies [13,25]. These volume differences were caused by ground measurements applying the geometric equation as explained in Figure 15 [25,51]. Indeed, similar magnitudes were observed between the two approaches; in fact, the largest and

smallest trees on the ground remained the same in the geometric reconstruction. Therefore, if one assumes that data from 3D reconstruction are able to determine a better estimate of CV, it is possible to better balance and manage certain agronomic practices such as variable-rate treatments, resulting in significant product savings. Such savings consequently translate into greater environmental and economic sustainability. CV showed a strong relationship with CA and TCSA, pointing out that vigour conditions are interconnected. From the cluster analysis, the vigour conditions were able to differentiate the real health status of each tree expressed by its production. Getting accurate data on plant vigour is an important condition to obtain the best growth pattern of the olive tree and to better manage the orchard [31].

## 5. Conclusions

This study was able to assess how the main growth parameters measured via a high-resolution remote platform and multispectral and RGB sensors processed on various GIS platforms can express the real field conditions and influence site-specific management of the olive grove. It was possible to verify that the new technologies available in precision agriculture allow to obtain various information on the health status of olive trees. Precisely, the UAV platform equipped with multispectral and RGB cameras was able to determine, through the GIS analysis, the main vegetative characteristics such as TCSA, CA and CV. They can be modified with the different agronomic practices to improve crop efficiency. UAV technology has demonstrated an excellent ability to efficiently produce spectral and geometric data of hundreds of agricultural trees at field level in a timely and accurate manner, offering a viable alternative to hard and inefficient field work by investigating the entire spatial variability of the orchard within minutes. In addition, the GIS platforms used were able to spatialize the collected point samples data, such as the nutritional ones. All geo-referenced information allows the creation of maps of orchard heterogeneity and the identification of incorrect growing conditions. This heterogeneity was expressed as spatial variability of different growth and production parameters. Knowing this variability is the key point for the creation of specific maps that allow the construction and use of accurate DSS systems for olive orchard management optimization. In this way, a site-specific management strategy can be applied to increase profitability by improving input utilization (fertilizers, pesticides, water, etc.) and field operations (pruning, spray application, irrigation, harvesting). The results obtained in this paper derive from the first study carried out in Sicily, a region of Italy that produces quality extra virgin olive oils. Further data and experimental results will be needed to validate these results.

**Author Contributions:** Conceptualization, P.C. and M.V.; methodology, P.C. and S.O.; validation, P.C., E.R. and S.O.; formal analysis, E.R.; investigation, P.C., S.O. and M.V.; resources, E.R. and P.C.; data curation, E.R.; writing—original draft preparation, E.R. and P.C.; writing—review and editing, M.V.; visualization, E.R. and M.V.; supervision, P.C. and S.O.; project administration, P.C. All authors have read and agreed to the published version of the manuscript.

**Funding:** This research received no external funding.

**Institutional Review Board Statement:** Not applicable.

**Informed Consent Statement:** Not applicable.

**Data Availability Statement:** Not applicable.

**Conflicts of Interest:** The authors declare no conflict of interest.

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
