# Peer review of "Evaluation of Multispectral Data Acquired from UAV Platform in Olive Orchard"

_horticulturae, doi:10.3390/horticulturae9020133_

Round 1

Reviewer 1 Report

 l.70-94 please split text into multiple paragraphs.

When you specify the term “vegetative parameters”, it would be useful to give some example. Also, sometimes is good to make distinction between the terms “parameters” and “traits”. For example, vegetation process of “growth” is influenced by environmental factors like precipitation and temperature, but in order to characterize this process we can also measure the biomass in leaf (which represents the plant trait), however, this trait also has some parameters like the leaf growth rate. In that sense, if it is of interest to measure some traits of olive trees, please consider whether this type of distinction of terms would be applicable in your case. In the following I give you some definitions in order to help, if necessary:

Trait: “morphological, biochemical, physiological, structural, phenological or behavioral characteristics that influence organism performance or fitness” (Nock et al, 2016). This is the intrinsic characteristic of the object.

Variable: A trait can be measured or estimated, and its value is called a “variable”. The variable will therefore depend on the method used to quantify the corresponding trait.

Parameter: A parameter is associated to a model describing a process. It is used to change the relationship between the inputs and the outputs. A parameter is specific to a model and cannot be measured directly (otherwhile it is a variable).

An example would be Fraction of Absorbed Photosynthetically Active Radiation, which is a trait, but is measured through PAR budget, or by ceptometer. Parameter in this case in the light use efficiency.

However, if you do not need this kind of nomenclature between the quantities, just make it clear in the introduction what is actually considered under the term parameters.

In section 2.5., please add more details or some functional diagram illustrating how the segmentation of  the tree crowns based on computed DSM was performed. In the way how it is presented it is not clear whether the area of the canopy was estimated from orthomosaics (representing original multispectral data, vegetation index or some combination) using some OBIA approach alone, or at the same time some DSM information was also used to ease the segmentation process (separating the tree crowns from the background). Also, later the DTM is used in eq. 3, but it is not described how  the DTM was obtained from DSM (did you performed some nonlinear filtering, like minimum filter, or some manual corrections, or something else). Related to this, in the text the term DEM is used, while in eq. 3 DTM. Why was important to have multispectral measurements for canopy characterization (or these were mainly used for NDVI – in that case, the relevance of the red edge band should also be considered, besides the NIR used for NDVI). Please consider to provide more details regarding the role of multispectral data.

lines 212-214 split into two sentences and make it more clear (the best would be to rewrite this section).

TSCA is not defined and introduced.

l.249, what is considered under the NDVI of each individual tree? Please provide more details in the text how this was computed (by averaging of NDVI of individual pixels in the tree canopy, or differently). This is also related to Figure 6, which needs to be better described in the text.

Line 274, what is DMS? L.276-277, how correlation coefficient was computed (what does it mean that data are not shown?) Why there was no diagram similar to the one in Figure 8, but with crown volume, instead of canopy area?

l. 284 “observed and estimated values for CA“, could you please refer to methodology how the observed CA values were determined. This is also some of information that is not clearly presented in the subsection 2.5.

In the discussion, please reference to tables in Figures presented in the results section, in order to support presented conclusions. Also, please clearly highlight the role of multispectral data, since the paper title suggests that it is one of the important elements of the study. Also, please try to clearly point out what are the main contributions of the paper, in terms of methodology, as well as the results and investigation related to the considered problems. In current form in needs to be improved in order to give the reader clear picture what are the advantages of the method presented in this study, and the conclusions regarding the capabilities of using described UAV platform in the task of determining the state of olive trees. In the discussion, pleas also add guidelines, how the same type of field work and the analysis should be used in some farming scenario (what should be measured, and what should be done after obtaining these information). From the perspective of applied research this is an important element (e.g. in the conclusion you mention the orchard heterogeneity, but in the results we have not seen any map illustrating this).

I propose that you carefully consider these and other comments and try to improve the manuscript content in order to better convey the main message of the conducted research. Besides that, I think that it is an interesting study, and that it should be a good candidate for possible publication, if these details are improved and better presented.

Nice work, good luck.

Author Response

Caro Revisore1,
grazie per i tuoi commenti e suggerimenti, sono convinto che serviranno a migliorare il manoscritto. 

Distinti saluti

Reviewer 2 Report

 Evaluation of multispectral data acquired from UAV platform in olive orchard

The current manuscript reports the application of MSI for characterization of olive plants using UAV. The work is interesting, and describes further application for MSI and UAV platforms, in addition to data analysis, since there are already several previous descriptions of such applications. Hence, the main novelty is the data processing and region of production in Italy.

 I believe that the work may contribute to the agricultural field, since the data is interesting and informative. I suggest the authors to enhance the discussion regarding two major topics:

 1) the aspects of chemical information that may be provided by spectral data, which may help readers to understand the information that may be provided. In this sense, the authors must include the spectral range of the camera, and compare to other HSI systems and applications for the chemical composition of vegetable samples used for direct consumption or oil extraction, such as olives. I hereby suggest some references that may help the authors, but it is just a suggestions, the authors may find others as suitable:

Early Detection and Quantification of Verticillium Wilt in Olive Using Hyperspectral and Thermal Imagery over Large Areas – https://doi.org/10.3390/rs70505584

High-resolution airborne hyperspectral and thermal imagery for early detection of Verticillium wilt of olive using fluorescence, temperature and narrow-band spectral indices - https://doi.org/10.1016/j.rse.2013.07.031

2) The authors have made a brief discussion regarding the nutrient composition (nitrogen) available to plants. I suggest the authors to try to improve this discussion, considering how the leaf composition may have affected the images acquired. Previous works have discussed this fact (only a suggestion, if the authors agree with the observation), and I think it is of utmost importance for agricultural field.

Influence of Plant Densities and Fertilization on Maize Grains by near Infrared Spectroscopy - https://doi.org/10.1080/00387010.2015.1076005

 Material and methods

Section 2.4. Image processing was performed manually or automatized by an algorithm? Please clarify the information.

 Results

The data presented in figure 6 shows very low correlation, and it could lead to even lower coefficient of prediction (r2). For instance, correlation of 0.42 presented in line 239 is very low. On the other hand, the results in figure 11 are much more promising than the others. I believe that the authors should be clearer about this low correlation, stating that in fact there is no correlation or it is low, and perhaps present possible reasons for this, instead of stating that it had good results. I believe it can help future researchers that may work in the same theme.

Minor comments

The text should be revised for minor typos and sentence construction. I just list a few mistakes, but there are others:

Line 144: It should be ‘total nitrogen content’

Author Response

Dear Reviewer2,
thank you for your comments and suggestions, I am convinced that they will serve to improve the manuscript. 

Best regards

Reviewer 3 Report

This paper investigates the ability of multispectral data acquired from a UAV platform to predict nutritional status, biometric characteristics, vegetative condition and production of olive orchards. According to the expression, organization, and experiment, I think there still have many problems in the paper.

1. This paper lacks technical innovation.

2. In the Introduction, the motivation and contribution of the method should be presented.

3. The advantages and limitations of related works are not well summarized, and should be expressed with more emphasis on the difference with the proposed method.

4. The overall flowchart of the proposed method should be presented.

5. The description of the method is not sound and detailed.

6. A comparison of the performance with other methods should be performed to demonstrate the effectiveness of the method.

7. More data and experimental results should be provided to support the conclusions.

8. Most of the references are outdated.

Author Response

Dear Reviewer3,
thank you for your comments and suggestions, I am sure they will help to improve the manuscript.
Kind regards 

Round 2

Reviewer 1 Report

The revised version contains details that were missing from the previous text. Therefore, it could be considered for possible publication by the journal.

Author Response

Dear Reviewer 1,

Thank you for your comments and suggestions of our manuscript.

PhD. Eliseo Roma

Reviewer 3 Report

There are still some problems in the paper as follows:

1. In the Introduction, the motivation and contribution of the method should be presented in detail.

2. The experimental results and conclusions on nutritional status, biometric characteristics, vegetative condition should be added.

3. CV* presented in Eq. (3), but there are no analytical and experimental results regarding this index.

4. The computation of CV should be given in detail.

Author Response

Dear Reviewer 3,

Thank you for your comments and suggestions to improve the manuscript's quality.
